# See Lung Cancer with an AI

**DOI:** 10.3390/cancers15041321

**Published:** 2023-02-19

**Authors:** Joanna Bidzińska, Edyta Szurowska

**Affiliations:** Second Department of Radiology, Medical University of Gdansk, 80-210 Gdańsk, Poland

**Keywords:** radiomics, lung cancer, artificial intelligence, lung cancer screening, precision, diagnostics

## Abstract

**Simple Summary:**

Lung cancer is the cause of many deaths that could have been avoided if the disease had been detected at an early stage. This is possible thanks to the lung cancer screening (LCS) program with the low-dose computed tomography (LDCT) of the chest. Due to the heavy workload on the healthcare system, shortages of specialists, and expensive equipment, new solutions are needed to support the work of the hospitals. One of the most promising is the use of artificial intelligence (AI). In this paper, we present promising results and discuss whether/why AI application in medicine, with an emphasis on lung cancer, is needed. It is speculated that thanks to an innovative AI solution many lives could be saved.

**Abstract:**

A lot has happened in the field of lung cancer screening in recent months. The ongoing discussion and documentation published by the scientific community and policymakers are of great importance to the entire European community and perhaps beyond. Lung cancer is the main worldwide killer. Low-dose computed tomography-based screening, together with smoking cessation, is the only tool to fight lung cancer, as it has already been proven in the United States of America but also European randomized controlled trials. Screening requires a lot of well-organized specialized work, but it can be supported by artificial intelligence (AI). Here we discuss whether and how to use AI for patients, radiologists, pulmonologists, thoracic surgeons, and all hospital staff supporting screening process benefits.

## 1. Introduction

Cancer is the second leading cause of death. According to GLOBOCAN 2020 estimates, cancer incidence reached more than 19 million cases and about 10 million cancer deaths worldwide [1].

Lung cancer was the most common cancer in men, reaching 1.43 million new cases diagnosed in 2020. Among women, there were 0.77 million cases which rank second and put breast cancer first, with 2.26 million cases (both genders) diagnosed in 2020. Lung cancer was the leading cause of cancer deaths with 1.8 million deaths (18%), 1.18 million among men, and 0.6 million among women. Cancer incidence estimates in 2040 are dramatic as the global burden of more than 28 million cases is expected, corresponding to a 47% increase from 2020 [1].

Importantly, the number of lung cancer deaths is close to the number of cases and still has a very poor prognosis with a 5-year survival rate of only about 13% in Europe [2]. This statistic derives from the late diagnosis, which is most often in stage IV. Only about 20% of lung cancer cases are stage I at the time of diagnosis. This dramatic situation did not change for decades.

In 1968 a publication describing the screening tests idea by Wilson & Jungner for WHO began the screening era and changed the situation of people at risk of silent and deadly diseases: ‘Screening is the presumptive identification of an unrecognized disease or defect by the application of tests, examinations, or other procedures which can be applied rapidly. Screening tests sort out well persons who probably have a disease from those who probably do not. A screening test is not intended to be diagnostic. Persons with positive or suspicious findings must be referred to their physicians for diagnosis and necessary treatment’ [3].

Concerning lung cancer, the main risk factor, among others, is tobacco smoking. It is responsible for more than 80% of lung cancer cases; moreover, it causes many other diseases including chronic obstructive pulmonary disease (COPD), cardiovascular disease (CVD), stroke, asthma, and other types of cancer.

The only way to win against this deadly disease is to detect it early when it is curable, and if possible, to treat by surgery, or sometimes, by stereotactic body radiation therapy (SBRT). This can be done by the broad implementation of primary and secondary prevention, ideally conducted simultaneously. Primary prevention emphasizes society education, prevention of smoking initiation, and help in quitting smoking.

Primary prevention is the most cost-effective intervention in the control of cancer. Secondary prevention through screening with the use of LDCT resulting in an early diagnosis of the disease is also a powerful tool to improve outcomes of patients from the high-risk group of developing lung cancer. This group consists of smokers with ≥20 pack-years, former smokers (≤15 years since quitting smoking), ≥55–74 years old, or ≥50–74 when there is an additional risk factor, such as history of lung cancer in the close family, exposition to asbestos, with slight variations between countries and studies [4,5]. One should keep in mind that former smokers remain at high risk of lung cancer for up to 25 years after stopping smoking [6].

Despite the existing evidence in randomized clinical trials on the effectiveness of LCS in reducing lung cancer mortality, so far screening has been introduced on a large scale only in the USA and to a limited extent in Canada, Poland, and the United Kingdom. Perhaps there is concern about LCS overburdening national health systems that already suffer from a shortage of specialist doctors, nurses, and other specialist medical personnel, and often insufficient equipment (especially CT scanners). Can AI facilitate the implementation of lung cancer screening in the world as a population-based program that must be introduced from a medical point of view?

In this review, we discuss the possibility and the need for AI use in everyday clinical work. We focus on its utility, with an emphasis on the radiomics use in lung cancer screening and lung cancer management.

## 2. Proof of Concept and Important Aspects of Lung Cancer Screening Implementation

### 2.1. Proof of Concept

Since the publication of the results of randomized controlled trials (RCT), the USA NLST study (2011) showed 20% lower mortality from lung cancer and was the only one that showed overall lower mortality in the LDCT than in the chest radiography screening group (which for some countries may be a requirement for implementation) [7]; the Italian MILD study (2019) [8], German LUSI (2020) [9], Dutch-Belgian NELSON study (2020), in which authors showed lower mortality rates in the LDCT study group by 26% in men and 39–61% in women [10], lung cancer screening has received even more attention, and many countries are working on introducing population-based lung cancer screening for people at high risk of the disease.

The 2020 meta-analysis of seven controlled RCTs (over 84,000 participants) showed a significant relative reduction in LC mortality by 17% and overall mortality by 4% in the LDCT screening group compared to the control group [11].

In the USA, on 5 February 2015, the Centers for Medicare and Medicaid Services (CMS) announced lung cancer screening coverage for people at high risk aged 66–77. Since then, almost 10 million Americans have had access to LDCT lung cancer screening.

Currently, CMS recommends lowering LCS initial age to 50/55 and smoking history requirements to 20-pack years from the previous 30-pack years. The coverage has been expanded nationwide to independent diagnostic facilities with the requirement for structured reporting [12].

### 2.2. Country-Specific Differences

In China, the statistics are very concerning as the reported LC mortality has increased by 465% percent in just 30 years, from 1975–2005 [13,14]. Since the early 2000s, multiple lung cancer screening studies have been conducted [14].

Interestingly in China, LCS is not limited to the “high-risk” population. LDCT is quite cheap, easy to access, and covered by healthcare insurance. In contrast to Western countries, in China, the results are quite surprising as the detected LCs concern a large part of young females and non-smokers. LC is detected in early-stage 0/I more often and usually, it is adenocarcinoma (ADC). There are government initiatives to implement a population-based screening program; however, in China no clear evidence of the ”high-risk” group exists [15].

In Europe (see Table 1), Croatia was the first country to implement a nationwide population-wide screening for lung cancer using low-dose chest computed tomography on 1 October 2020. The program is financed by the Croatian Health Insurance Fund. Pilot studies are running in Poland (WWRP 2020–2023), Italy, Great Britain, Norway, Spain, France (CASCADE study—solely female smokers), and Germany [16].

Based on the last UK NSC review from June 2022, targeted screening for lung cancer with integrated smoking cessation service provision is recommended for people aged 55 to 74 identified as being at high risk of lung cancer. Targeted Lung Health Checks (TLHC) are an effective starting point for LCS implementation in England [17].
cancers-15-01321-t001_Table 1Table 1Lung cancer screening in European countries according to the first edition of the Interactive Map of Lung Cancer Screening. According to the data collected by The Lung Cancer Policy Network (LCPN) 2022. Status from the LCPN data [18].CountryStatusAlbania, Bosnia and Herzegovina, Bulgaria, Cyprus, Iceland, Latvia, Lithuania, Malta, Moldova, Montenegro, North Macedonia, Portugal, SloveniaUnknown status. No further information available.Belarus, Norway, SwitzerlandImplementation research. No further information available.AustriaImplementation research. The Austrian Society of Pneumology and the Austrian Radiological Society recommended the implementation of LDCT screening and formed a taskforce focused on this in 2018. In February 2021, it announced plans to implement future pilots looking at the feasibility of LDCT screening in Austria. No further information is available.BelgiumImplementation research. Belgium participated in a large clinical effectiveness RCT (NELSON), and results were reported in 2020. The decision to implement LCS is governed by the Flemish, Walloon, or Brussels’ Ministers of Health. Funding was recently approved for a pilot in the Flemish region.CroatiaImplementation roll-out. Croatia launched a national LDCT LCS in October 2020, as the first country in Europe. It is financed by the Croatian Health Insurance Fund. First term will conclude in 2023 before renewal.Czech RepublicImplementation roll-out. In January 2022, a five-year national LCS was initiated by the Czech Ministry of Health. It is fully covered by public health insurance. The anticipated completion is 2026. It is expected to be renewed.DenmarkUnknown status. Denmark previously conducted a large clinical effectiveness RCT (the DLCST), which ended in 2015. In January 2021, a proposal was submitted to the Danish National Board of Health (NBH) to introduce a screening pilot in Denmark. Currently, the NBH and Ministry of Health are considering the proposal and possible funding of a pilot as part of a HTA evaluation, which would serve as the first step towards national implementation.EstoniaImplementation research. An ongoing regional pilot in Tartu aims to assess the feasibility of introducing a national LCS. The pilot builds on earlier implementation research, including a study conducted by the National Institute for Health Development (TAI).FinlandUnknown status. In November 2021, a working group appointed by the Finnish Medical Association Duodecim, the Finnish Lung Doctors’ Association, and the Finnish Society for Oncology reviewed its clinical guidelines for lung cancer. It concluded that further research was required around the optimal eligibility criteria, interval, and cost-effectiveness of LDCT LCS before its implementation.FranceImplementation research. Pilots ongoing in France, with funding from public and private sources. In February 2022, the High Authority for Health (HAS) announced it will carry out an update to the 2016 evaluation of the evidence for LDCT LCS, and nationally-funded implementation pilots will begin.GermanyImplementation research. Several clinical effectiveness trials and economic evaluations have previously taken place in Germany. Two prospective evaluations performed by the Federal Office for Radiation Protection and the Institute for Quality and Efficiency in Health Care (IQWiG) were supportive of LCS in 2020, and this must be approved by The Federal Joint Committee (G-BA). Several implementation trials are ongoing (HANSE and the 4-IN-THE-LUNG-RUN).GreeceUnknown status. As of 2021, there was no official recommendation from the Greek Ministry of Health regarding LDCT LCS. The development of a National Lung Cancer Control Strategy which includes early detection was announced. On a private level, there are implementation studies around screening programmes in a select few hospitals in Athens, Thessaloniki, and Crete.HungaryImplementation research. The first phase of a national LDCT lung cancer screening pilot program (HUNCHEST) was completed in 2018. The second phase is currently ongoing and expected to end in 2022.IrelandUnknown status. There are a few small-scale opportunistic screening programmes for lung cancer, but these are exclusive to private hospitals and not endorsed by the NCCP or National Screening Service.ItalyFormal commitment. In Italy, the Ministry of Health has committed to a national programme of implementation, with a national pilot study being conducted in several centers. Italy also led several large RCTs investigating the clinical effectiveness of LDCT screening.LuxembourgUnknown status. In 2020, it was reported that preliminary discussions for the implementation of an LDCT LCS had begun with the National healthcare system and National Health Insurance Fund.NetherlandsImplementation research. The Netherlands participated in a large clinical effectiveness RCT (NELSON); results were reported in 2020. Recruitment recently began for the Dutch arm of the 4-IN-THE-LUNG-RUN trial, which is due to end in 2024. The government has yet to support the implementation of LDCT screening.United KingdomFormal commitment. The UK National Screening Committee (UK NSC) recently reviewed its recommendations for lung cancer screening, the outcome of which was announced in September 2022. The UK NSC now recommends LDCT screening for current and former smokers aged 55–74. England and Scotland are currently engaged in implementation research (e.g., LUNGSCOT study and the Targeted Lung Health Check pilot programme).PolandImplementation roll-out. Poland initiated pilot studies for the early detection of lung cancer via LDCT screening in 2008. These pilots were introduced independently in Szczecin, Gdańsk, Poznań, and Warsaw. In 2020, a national programme (the WWRP) began. The three-year centrally administered programme is being implemented. The country is divided in six macroregions with Centers of Excellence and cooperation sites.RomaniaFormal commitment. The latest NCCP has pledged that the Ministry of Health and the National Institute of Public Health will develop a national pilot programme between 2023 and 2025. The proposed target population is current and former smokers aged 50–80 who will be offered annual LDCT screening.SerbiaImplementation research. The first screening pilot programme in Serbia is currently ongoing in the Autonomous Province of Vojvodina (APV). Initially planned as a region-wide pilot, due to the COVID-19 pandemic, it is only taking place in the South Bačka district.SlovakiaImplementation research. A Ministry of Health taskforce is currently developing a plan for a national LDCT lung cancer screening programme in Slovakia.SpainImplementation research. There is no national LCS program in Spain. There have been some implementation studies. Currently, there are plans for a national pilot (CASSANDRA) led by the Spanish Society of Pneumology and Thoracic Surgery (SEPAR). This is due to begin in 2022.SwedenImplementation research. A regional cancer center was commissioned to deliver a regional lung cancer screening pilot that targets women in Stockholm. If successful, findings will be used to inform the development of a national organized LDCT screening programme, in which men would also be eligible to participate.


### 2.3. Obstacles to Overcome

#### 2.3.1. False Positive Results

Apart from all the benefits of the intervention, we must remember the challenges in LCS, such as high false-positive rates (FPR) (NELSON showed that the use of nodule VDT results in fewer FPs), radiation (detection-over-induction ratio), efficient work-flow organization and its standardization, and finally overall quality assurance and its monitoring.

Several factors could influence FPR in LDCT LCS including patient-specific and site/radiologist-specific factors. Less experienced radiologists are more prone to have a higher false-positive rate [19,20,21].

#### 2.3.2. Political Level and Cost-Effectiveness

In our opinion, the main obstacle to the implementation of population-based LC screening is an ongoing debate concerning the cost-effectiveness of the intervention in the political environment.

Modeling studies have shown that LDCT screening is cost-effective, but the difficulty lies in its implementation in diverse international healthcare systems [22,23,24,25,26,27,28]. The cost of discoveries and clinical application of approved innovative therapies such as targeted drugs and immunotherapies should also be considered. As reported by Van Meerbeeck and Franck, after personal contact with M. Neyt, according to recent data from the Belgian HTA KCE Institute, systemic treatment expenditure at stage 4 LC has tripled since the introduction of checkpoint inhibitors (M. Neyt, personal communication, 2020) [16].

Lung cancer treatment is a huge medical and financial challenge. There are large differences in cost/quality-adjusted life years (QALYs) depending on the treatment modality of the disease. This in turn depends on the stage of the disease at the diagnosis. For example, the incremental cost-effectiveness ratio (ICER) of a lobectomy in the early stage is 10,000 €/QALY, of chemo-radiotherapy in locally advanced LC, is 16,000 €/QALY, and of first-line chemotherapy in advanced disease is 20,000 €/QALY [29]. The one QALY reaches 25,000 € for targeted drugs and a dizzying 100,000 €/QALY for immunotherapy [16,30].

With that in mind, that is, cost-effectiveness, treatment costs depending on the disease stage, and mortality rates, there is a broad agreement among scientific experts and clinicians that an effective screening program should be immediately implemented in Europe. In 2017, the European position statement on lung cancer screening was published [31].

### 2.4. Hope or Hype from the European Commission?

Europe’s Beating Cancer Plan of the European Commission has been part of the European Health Union since September 2020. It was launched in 2021 to deliver integrated, standardized cancer prevention, treatment, and care. In 2022, work programs include activities focusing on cancer prevention and diagnosis [32,33].

In the Report on Strengthening Europe in the Fight Against Cancer from 7 January 2022, the Special Committee on Beating Cancer mentions, in point 55, the update of the 2003 Council Recommendation on cancer screening where new screening tests will be considered and should be evaluated by experts. Furthermore, it calls on the European Commission to develop research guidelines for the evaluation and potential inclusion of new cancer screening programs including lung, ovarian, prostate, and stomach cancer. The role of artificial intelligence (AI) as a part of the 2022 Council Recommendation is also mentioned [34].

In March 2022, Science Advice for Policy by European Academies (SAPEA), part of the Scientific Advice Mechanism of the European Commission (SAM), which supports the College of Commissioners with evidence-based, timely, independent scientific advice for policy making, published the report with the advice for the introduction of lung cancer (and prostate cancer) screening program in EU member states as a life-saving intervention [35].

During the preparation of this work, on 20 September 2022, after about two decades since the first recommendations, the European Commission unveiled its proposal for a “Council recommendation on strengthening prevention through an early detection: A new EU approach on cancer screening” to boost cancer screenings in the EU, recommending that three new types of cancer should be screened for more systematically, including LCS [36].

This document was highly anticipated and the whole community pinned hopes on it and thought it would be a turning point for LCS in Europe. Unfortunately, the European Commission did not sufficiently emphasize the need to introduce lung cancer screening despite the existence of irrefutable scientific evidence, thus failing the lung cancer medical and patient communities.

Lung Cancer Policy Network, a global multidisciplinary alliance consisting of more than 50 experts from across the lung cancer community, that aims to make lung cancer an international policy priority, on 22 October 2022, submitted to the EU a document ‘Proposed amendments to the EU Commission draft recommendation on cancer screening’.

The document created by clinicians, researchers, patient organizations, and industry partners underlines that ‘aspects of the proposal could be strengthened to fully reflect the wealth of evidence available to guide the implementation of targeted LDCT screening for lung cancer in EU [37,38,39].

## 3. What about Tobacco?

### 3.1. Horrific Statistics

Smoking is one of the biggest health problems worldwide. The World Health Organization (WHO) estimates that in 2020, around 1.3 billion people worldwide suffer from tobacco addiction; this is about 1/4 of the population, 36.7% of all men, and 7.8% of the world’s women.

Tobacco kills more than 8 million people each year. Around 1.2 million deaths are the result of non-smokers being exposed to second-hand smoke [40].

The largest percentage of smokers is recorded in Asian countries: China, Indonesia, and India, where 400 million men admit to smoking. The largest numbers of women who smoke cigarettes are in the United States, India, and China (13–17 million smokers).

Tobacco smoking decreases significantly from year to year, on average by 2–4%. In Brazil over the last 29 years, the number of heavy smokers has decreased by about 60% [40].

According to Eurobarometer, one of the European countries where the most cigarettes are smoked per day is Austria (19.8 cigarettes/day). In Cyprus and Greece, an average is 19.5 cigarettes per day. Every tenth European smokes more than 21 cigarettes per day. In Poland, smokers reach about 15.6 cigarettes per day. The best result in Europe was reported in Sweden, with the lowest indicator of 9.9 cigarettes/day [40].

In the European Union, smoking kills an average of 700,000 people a year. These deaths could have been avoided [40].

### 3.2. Beyond the Tobacco

Moreover, WHO in the ‘Report on the global tobacco epidemic 2021: addressing new and emerging products’ warns that in recent years, newer and emerging nicotine and tobacco products, such as electronic nicotine delivery systems (ENDS), have proliferated in many markets. The tobacco industry implies that ENDS are safer because they do not contain tobacco, but ENDS continues to grow the industry’s customer base, attracting younger users [40,41].

## 4. The Need for Immediate Action

Statistics and facts call for the need for immediate LDCT lung cancer screening implementation and anti-tobacco European actions.

To put effective solutions in place, at this point the inevitable questions are:Will this put a strain on healthcare systems with staff and equipment shortages in some countries?Could an artificial intelligence application in healthcare solve some of these problems?

Hospitals are on the front line of taking care of smoking-related diseases. Tobacco smoking also puts a great financial and social burden on hospitals and the loss of productivity attributable to morbidity and mortality.

Concerning the influence of the implementation of populational lung cancer screening, in the example of the USA, or other countries where LC screening is live, we can say that it is already known, that well-planned and organized screening is efficient and beneficial both in terms of patient health and the costs of possible treatment of cancer, but also other tobacco-related diseases.

Treating advanced lung cancer is very expensive, and secondhand smoke exposure can lead to additional hospitalizations and interventions. Evidence on the tobacco use influence on economic costs is still too low but luckily is growing.

Modern computed tomography machines are expensive. This may be a barrier to the introduction of screening tests where many scans are performed, ideally in many trained sites. However, there is an improvement in the situation on the horizon due to the possibility of financing the purchase of equipment with funding from national and international grants.

Radiologists and clinicians are an integral part of patient care. Currently, many hospitals face the problem of radiologist and other staff shortages. In the UK, the last census of the Royal College of Radiologists showed a shortfall of 29%, putting doctors under huge pressure [42].

Companies assuring remote descriptions of CT examinations within 24 h appear on the market. High hopes in the support of radiologists are pinned on the use of artificial intelligence and the first tools are already available. They allow for the triage of examinations, recognize changes in scans, and estimate their features and nature, thus shortening the working time of the radiologist, who is responsible for the final assessment and decision.

## 5. Artificial Intelligence in Healthcare with a Focus on Oncology

Artificial intelligence, after decades of research, is beginning to enter everyday clinical practice. Among AI technologies, the most recognizable are the traditional Machine Learning (ML) and Deep Learning (DL) algorithms created with the use of complex problem-solving neural network models (DDNs—deep neural networks, CNNs—convolutional neural networks, RNNs—recurrent neural networks) [43] and radiomics, which allow extraction of a set of features from a medical image and their automated classification into a predefined group [44].

On 19 February 2022, the European Commission launched White Paper on ‘Artificial Intelligence—a European approach to excellence and trust’ to promote AI and to address the risks associated with its specific uses. The paper contains the set of fundamental rights as well as rules for assurance of safety and equal protection for the harmed people.

AI as a strategic technology offers many benefits for society; however, the technology must be human-oriented and most of all ethical. AI offers the improvement of productivity and well-being by helping to solve difficult societal challenges including health protection [14,45].

The White Paper is directly connected with the ‘Report on the safety and liability implications of Artificial Intelligence, the Internet of Things and Robotics’ (19 February 2020). The EU launched a consultation to allow a comprehensive dialogue with all concerned parties (Member States’ civil society, industry, academics) concerning the approach to the AI in the field of research and innovation and investments and further regulations. The challenges to the existing liability rules have also been identified [45,46].

On 28 September 2022, the EU Commission delivered the Proposal for an Artificial Intelligence Liability Directive (AILD). The AILD ensures, among others, that justified claims are not hindered [45,47].

For many years, imaging has been used in cancer research and diagnosis. New knowledge in the field of cancer biology and continuous technological progress has resulted in the use of multiple imaging modalities, including MRI, CT, USG, PET, and SPECT for screening, staging, therapy/radiotherapy, surgery, assessment of response to therapy, prognosis, and relapse. The addition of AI resulted in remarkable advances in cancer treatment. Clinicians, radiologists, and other professionals have gained a trusted assistant to relieve workload and potentially shorten clinical decision-making time for a cancer patient.

In 1996 the National Cancer Institute (NCI) established the Diagnostic Imaging Program, the name of which was changed to Biomedical Imaging Program and eventually to the Cancer Imaging Program (CIP) in 2003. The NCI CIP’s mission is to support advances in in vivo medical imaging in basic and applied research. In addition, it promotes imaging in clinical trials for the benefit of patients. CIP initiatives include, among many others, Cancer Imaging Archive (https://www.cancerimagingarchive.net, accessed on 10 December 2022) and Lung Image Database for Imaging Research (LIDC). LIDC is composed of US-based academic institutions that aim to create an international image research resource for the development, training, and testing of CAD techniques for the detection of lung nodules on CT scans [48,49].

There is a growing variety of possibilities for using AI in the management of people at risk and cancer patients. Among recent advances, we can find methods supported by AI to early detect, diagnose, classify, and monitor the development and progression of the disease or its response to therapy. For example, it is possible to exploit radiographic imaging, cancer genome (TheCancerGenomeAtlas (TCGA) [50]), and medical records databases, to analyze the biomarkers and tumor microenvironment [51,52,53,54].

Claudio Luchini and Antonio Pea from the University and Hospital Trust of Verona retrieved all AI-based devices that have obtained Federal Drug Administration (FDA) approval in oncology-related fields. They extracted all potential data by searching FDA official databases and found 71 AI-associated or AI-associable devices approved by the FDA [55] (https://www.fda.gov/downloads/medicaldevices/deviceregulationandguidance/guidancedocuments/ucm514737.pdf, https://www.fda.gov/media/145022/download, https://www.accessdata.fda.gov/scripts/cdrh/cfdocs/cfPMN/denovo.cfm, accessed on 10 December 2022).

The Authors found out that 54.9% of oncology-related FDA-approved devices are from the cancer radiology field, and interestingly, >80% of the devices concern the cancer diagnostics area. They can be exploited to study solid malignancies—33.8% (cancer in general) where 31% of the devices are related to breast cancer and lung and prostate cancer (8.5% each) [55]. Still, the methodology used for the creation of the devices must be kept in mind, especially considering the number of cases used for the algorithms training and validation studies. Particular attention should be paid to the data that is used to learn and validate AI systems because their quality has a huge impact on the creation of trustworthy AI systems.

## 6. Radiomics in Lung Cancer Screening, Diagnostics, and Prognostication

### 6.1. Basic Concept

In 2012, the concept of radiomics has been proposed for the first time [56]. A simple explanation is provided in the short video produced by Alpha Grid on behalf of the Maestro Clinic, Netherlands on the decision-supporting system https://www.youtube.com/watch?v=Tq980GEVP0Y (accessed on 10 December 2022) [57]. Radiomics treats images as quantitative and minable data associated with clinical events [57]. The radiomic workflow shown in Table 2 consists of the subsequent steps and should be planned and performed carefully [58,59,60,61].

Even though radiomics is a powerful tool for differentiation of the benign and malignant tumors, it has technical limitations in assessing cancer genetics, staging, or treatment response prediction. For example, due to the different texture sensitivity, image acquisition mode and type of processing can influence the given radiomic feature repeatability and reproducibility [62,63,64]. In the next subsections, we briefly exemplify AI’s potential utility.

### 6.2. Radiomics in Lung Cancer Screening

Considering dramatic lung cancer morbidity and mortality statistics, new methods of early detection (e.g., biomarkers, radiomic signatures) hopefully will be validated in the nearest future and used in populational screening programs in the whole of Europe together with the LDCT. Lung cancer is an asymptomatic disease until the late development stage when it is not effectively treatable and in consequence, the survival rate is very low. People at risk of the disease should be systematically screened.

Non-small lung cancer (NSCLC) is one of the most studied diseases regarding radiomics utility for personalized medicine applications [65]. As imaging is widely used, radiomics can have a great impact, and together with the LDCT be a game changer.

Several studies confirmed the utility of radiomics in the classification of lung nodules, their histopathological recognition, and invasiveness assessment. Commercial and open-source programs can speed up the radiological workflow by using automatic solutions [66].

Among computer-aided detection systems, many algorithms are useful for pulmonary nodule detection. One of the most known systems for evaluating radiological examinations based on artificial intelligence is AI-RAD Companion developed by Siemens Healthcare which allows three-dimensional segmentation of the lungs. AI-RAD Companion also enables volume quantification of lungs, lobes, lesions, thoracic aorta, and coronary artery calcification. The artificial intelligence algorithms used in the AI-RAD Companion automatically use image data. The AI-RAD Companion allows the automation of routine workflows of repetitive tasks. Streamlined daily work allows specialists to save time and make fewer mistakes (https://www.siemens-healthineers.com/digital-health-solutions/digital-solutions-overview/clinical-decision-support/ai-rad-companion, accessed on 19 January 2023) [67]. After the automatic assessment of the scan by AI-RAD Companion, it assists the user in the automatic interpretation of the data. In the last step, the automatically generated result can be verified, delivered, sent, and saved to the report (https://grand-challenge.org/aiforradiology/product/siemens-rad-companion-chest-ct/, accessed on 19 January 20223) [68].

Another solution had been developed by the Aidence company. Aidence provides AI-powered clinical applications for the management of lung cancer including two tools:Veye Lung Nodules—CE-certified automated lung nodule management assistant integrated into the radiological workflow. It can be used for automated lung nodule detection and quantification on chest CT scans.Veye Reporting—the interactive solution for lung nodule reporting allows for the generation of standardized quality reporting.

Both Aidence tools can be applied in lung cancer screening but also can be used in clinical work. Aidence solutions are broadly used in the UK hospitals that are part of the Targeted Lung Health Check Program (Lung Cancer Screening) and across Europe in at least eight countries in clinical practice. In January 2022, Aidence was acquired by RadNet, a US leader in diagnostic imaging services (https://www.aidence.com, https://www.bir.org.uk/media/477315/lung_cancer_and_ai_final_updated_v2_150622.pdf, https://www.radnet.com, accessed on 19 January 2023) [69,70,71].

Aidoc’s pulmonary nodules solution is a triage and notification software indicated for use in the analysis of CT images. It flags and communicates suspected positive findings of pulmonary nodules (https://grand-challenge.org/aiforradiology/product/aidoc-pulmonary-nodules/, https://www.aidoc.com/radiology-ai/, accessed on 19 January 2023) [72,73].

Veolity is a reading platform dedicated to the lung cancer screening program for nodule detection, segmentation, and quantification. It allows nodule comparison and report generation. It automatically marks regions that are suggestive of solid pulmonary nodules using a CAD algorithm (https://grand-challenge.org/aiforradiology/product/mevis-veolity/, https://www.veolity.com, accessed on 19 January 2023) [74,75].

InferRead CT Lung is a solution for lung cancer screening. It recognizes and determines the characteristics of suspected lung nodules, provides nodules localization, size, density, and malignancy rate and report generation (https://grand-challenge.org/aiforradiology/product/infervision-ct-lung/, https://global.infervision.com, accessed on 19 January 2023) [76,77].

JLD-01K is based on the deep learning model—Convolutional Neural Network (CNN). It detects and measures the volume and diameter of nodules on CT pulmonary images. The categorization is made according to the Lung-RADS. In addition, lungs and nodules can be visualized and displayed through 3D views (https://grand-challenge.org/aiforradiology/product/jlk-inc-jld-01k/, accessed on 19 January 2023) [78].

The LCS+ tool from Coreline Soft can be used for the analysis of the three main lung diseases: lung cancer, COPD, and coronary artery calcification, based on different types of CT scans. It enables assessment of nodule volume quantification, VDT, and Lung-RADS score (https://grand-challenge.org/aiforradiology/product/coreline-soft-aview-lcs/, https://www.corelinesoft.com/en/lcs/, accessed on 19 January 2023) [79,80,81,82,83].

Taking into account how many solutions have become available, it is important that, before buying the system, it is usually possible to test it on cases, for example, with the Contexflow (https://calendly.com/contextflow-js/contextflow-demo?month=2023-02, accessed on 19 January 2023) or ask for a demo [84].

Contexflow is creating AI tools for lung nodules assessment. During the last RSNA Congress (2022), the company presented an outstanding application allowing for the automatic primary nodule malignancy assessment. It showed that a ‘computer-assisted diagnosis software improved risk classification from chest CTs of screening and incidentally detected lung nodules compared with Lung-RADS’. The authors claim that ‘these results suggest the generalizability and potential clinical impact of a tool that is straightforward to implement in practice’ [66,68]. The Contexflow company videos are available here: ECR 2022: interviews from working radiologists about the value context flow can provide in terms of integration and diagnostic support: https://www.youtube.com/watch?v=zTR0oZEto3g, https://www.youtube.com/watch?v=VR7uNZW536E, (accessed on 19 January 2023).

Forte et al., in a recent systematic review and meta-analysis of deep learning algorithms for lung cancer diagnosis, found that the pooled sensitivity and specificity of DL networks for the diagnosis of lung cancer were 93% and 68%, respectively. Current data concerning DL-based CAD tools will play an important role in LCS, but many improvements can still be made [85].

### 6.3. Detection, Diagnosis, Staging

Diagnosis and especially fast and proper diagnosis are often a factor that determines further life expectancy. Due to the complex cancer biology/histology, it is of great value to develop the tools which can help clinicians and pathologists to identify the disease better and faster.

Previously in the 90s, CT-based radiomic features have been used to classify a pulmonary nodule as benign or malignant [86].

Antonio Brunetti et al. developed a workflow that allows for efficient classification of the histological subtype in lung nodules to discriminate between lung adenocarcinomas and other cancer types. The radiomic signature with a reduced set of radiomic features has been used. At the same time, the Authors underline the importance of the reproducibility of radiomic studies on external validation data [87].

High-precision detection of lung nodules is challenging. To address the problem, many groups aim to develop algorithms and methods for their automatic detection which could greatly improve work efficiency and accuracy rate and help specialists in the diagnostic process.

Among new algorithms for the detection of pulmonary nodules are a polygonal approximation and hybrid features from CT images [88] and a neuro-evolutionary scheme [89]. For feature extraction (shape, intensity, texture) and nodule candidate classification, one can use the incorporation of 3D tensor filtering with local image feature analysis [90]. Another one of the several new methods for feature extraction and nodule candidate classification published recently is spectral analysis with the use of the optimal fractional S-transform that is applied to transform raw images from the spatial to time–frequency. After a few steps, nodule candidates are detected using rule-based and threshold algorithms in the Teager–Kaiser energy image. The proposed method exhibits a sensitivity of 97.87% and can efficiently reduce the number of false positives [91].

Zhao et al. propose a pulmonary nodule detection method based on deep learning. In the first stage, the multiscale features, and Faster R-CNN, are combined to improve the detection of small-sized lung nodules. To reduce false-positive nodules, in the second stage, a 3D convolutional neural network based on multiscale fusion was designed. The Authors report that the nodule detection model has achieved a sensitivity of 98.6%, and the proposed method reached a sensitivity of 90.5%. It has been shown that the detection method of pulmonary nodules based on multiscale fusion has a higher detection rate for small nodules. It improves the classification performance of true- and false-positive pulmonary nodules. It outperforms other published methods and can help specialists in the diagnostic process [92].

Lin J. et al. developed a new 3D framework IR-UNet + + for automatic pulmonary nodule detection based on three steps. First is the combination of the Inception Net and ResNet as building blocks. Second, the squeeze-and-excitation structure is introduced into these building blocks for more efficient feature extraction. In the last step, two short skip pathways are redesigned based on the U-shaped network. The developed model has been shown to perform better than other lung nodule detection methods. Achieved sensitivity is 1 FP/scan, 4 FPs/scan, and 8 FPs/scan which is 90.13%, 94.77%, and 95.78%, respectively [93].

Sethy et al. developed a hybrid network for the categorization of lung histopathology images, by combining AlexNet, wavelet, and support vector machines. For pulmonary nodules classification, they feed the integrated discrete wavelet transform (DWT) coefficients and AlexNet deep features into linear support vector machines (SVMs). To train and test SVM classifiers, the 5000 digital histopathology image datasets have been used. The images were divided into three categories: normal, adenocarcinoma, and squamous carcinoma cells. An accuracy of 99.3% has been achieved using a 10-fold cross-validation method and AUC of 0.99 in classification [94].

Gugulothu et al. created a novel algorithm for early diagnosis of lung cancer detection and classification. The Adaptive Mode Ostu Binarization technique is used for the Lung Volumes isextortedas of the image with the extracted lung regions pre-processing. After the detection, segmentation takes place utilizing Geodesic Fuzzy C-Means Clustering Segmentation Algorithm. After feature extraction, nodules are classified by Logarithmic Layer Xception Neural Network Classifier. The results show enhanced classification accuracy vs prevailing techniques [95].

Lekshmi Thattaamuriyil Padmakumari et al. studied an interesting issue i.e., retrospective discrimination of LC from tuberculosis using chest CT radiomics and have shown that radiomics may be a non-invasive imaging tool for differentiating LC; however, future prospective studies are needed to validate these initial findings [96].

Despite all the knowledge we have today, there is still a great need to improve existing algorithms for lung cancer detection, diagnosis, and stage of disease assessment.

### 6.4. Prognosis/Prediction

Additionally, radiomics could be efficiently applied for the assessment of the response to treatment. Ch. et al. have studied if peritumoral, intratumoral, or combined CT radiomic features can predict chemotherapy response in NSCLC and found out that both are efficient using machine learning models, and combined features from two peritumoral regions yielded better results [97].

As early as 2014, Aerts et al. using CT images assessed the prognostic values of 440 shape- and intensity-based and textural features and identified features that were predictive of patients’ survival. Moreover, the prognostic value of features was validated also in the non-lung cancer cohort, and the potential use of radiomic features in outcome prediction and describing intratumoral heterogeneity has been confirmed. In addition, this study has shown that prognostic ability may be transferred from one disease type to another [98]. However, some studies suggest the cancer-specificity of selected radiomic features; therefore, it must be taken with caution.

In CT imaging, Aerts et al. (2014) found that radiomic features related to the shape and wavelet features describing the heterogeneous phenotype of lung tumors were found to be significantly associated with the cell cycle pathway, suggesting that highly proliferative tumors demonstrate complex imaging patterns. Moreover, various biological mechanisms may be described by different radiomic features as the features were found to be related to different biological gene sets, including DNA recombination and regulation of DNA metabolic processes [98].

Radiomic features could also be exploited to predict the tumor’s metastatic potential. There is great potential for their use in a computational model to aid lung cancer histopathological subtype diagnosis as a “virtual biopsy” and metastatic prediction for therapy decision support [95,96,99,100].

Digital pathology could be used to predict patient prognosis and response to treatment. Straight linking of specific pathological features with survival outcomes is a great challenge. Could AI complement this assessment? Could radiomics and digital pathology be interconnected?

This year for the first time Zhou et al. applied multiple machine learning algorithms, gene expression online databases, and in vitro experiments to show a potential biomarker for lung adenocarcinoma. They used three datasets from the omnibus database and applied R software to screen differentially expressed genes together with the immune cell infiltrates analysis. Further steps were performed with the use of multiple machine learning algorithms (least absolute shrinkage and selection operator, support vector machine-recursive feature elimination). The in vitro experiments have also been performed. The biomarker—Topoisomerase II alpha—is overexpressed in lung cancer and associated with a poor prognosis [101].

Patients with the EGFR mutation have a bad prognosis. Early prediction of disease progression can help to manage the treatment course. A deep learning method using CT characteristics and clinical data to predict progression-free survival in patients with NSCLC after EGFR-TKI treatment has been developed. The combinational DeepSurv model performed better than a model based solely on clinical data, and PFS can help in the prediction of tumor progression [102].

Shimada et al. have shown that CT-based radiomics with AI in predicting pathological lymph node metastasis in patients with clinical stage 0-IA non-small cell lung cancer may have broad applications such as guiding individualized surgical approaches and post-operative treatment. They used AI software Beta Version (Fujifilm Corporation, Japan), 39 AI imaging factors, including 17 factors from the AI ground-glass nodule analysis, and 22 radiomic features from nodule characterization analysis [103].

The prognostic role of the stromal components in small-size tumors with lepidic components remains a challenge. Using a machine learning algorithm, the Authors analyzed the prognostic impact of each tumor component. They were able to stratify the post-operative prognosis of surgically resected stage IA adenocarcinomas and have shown that this method might help in the selection of patients with a high risk of recurrence [104].

Pan Z et al. developed an Optimized early Warning model for Lung cancer risk (OWL) using the XGBoost algorithm. They used a machine learning technique with a wide list of questionnaire-based predictors and obtained a high degree of predictive accuracy and robustness with clinical utility that can aid in screening individuals with high risks of lung cancer [105].

For the prediction of invasiveness of lung cancer based on preoperative [^18^F]fluorodeoxyglucose positron emission tomography and CT radiomic features, seven machine learning models were developed and validated. Radiomics features were extracted with the PyRadiomics package. The developed machine learning model was able to predict pathological highly invasive lung cancer with high discriminative ability and stability. It could be useful in quantitative risk assessment [106].

In the recent review of Ge et al., the Author provides an overview of the commonly used radiomic feature selection methods and predictive models. They compare the limitations of various methods in clinical applications. The sources of uncertainty are also presented. The impact of radiomic features, models, and methods on the integrity of radiomics studies should not be omitted [107].

## 7. Future or Current Directions?

Apart from radiomics, other AI tools are also used in lung cancer detection, diagnostics, and prognostication. They may and should complement each other to obtain better clinical results. Precise disease assessment is the clue for the proper clinical staging, histopathological analysis results, genomic features of the disease identification, and finally, treatment of lung cancer. AI, an algorithm for the prediction and classification of objects with the use of the existing data, using, for example, logistic regression, can simplify and optimize these processes. Among them, we can list machine learning which includes decision trees, support vector machines (SVMs), and Bayesian networks. Further neural networks, deep learning, and convolutional neural networks can also be exploited [108,109].

For diagnosis, one can assess the nodule’s histopathological features with the tool of digital pathology—whole slide imaging (WSI) which allows recognition, segmentation, classification, and markers scoring. The slide scanner transforms glass slides into digital images which are stored on the server with access for the pathologists to share expertise [110,111,112,113]. It has been shown that the WSI model can outperform the pathologist in the analysis of H&E-stained slides [114]. Moreover, it is possible to predict specific gene mutations (e.g., tumor-specific receptors) from the stained WSI slides; consequently, it enables the assessment of the treatment response and the prognosis of patients. Interestingly, expression levels seem to be related to the observed morphological features [115]. AI can also count the immune cells on slides stained for markers that are known predictor markers, such as PD-L1 for immunotherapy response [116,117,118].

The WSI can also be used in the analysis of cytological slides. In digitalized cytology slides, the focus function is simulated through the Z-stack function and multiple layers of a different focus. What is important is the multi-potency of this method in the means of the material which can be used for the analysis. The cytological sample derived from lung cancer patients can be taken from pleural effusion, tissue aspiration, lymph node aspiration, or endobronchial ultrasound-guided fine-needle aspiration of mediastinal lymph nodes [119,120].

A combination of AI methods can help in the treatment choice including drug, surgery, and/or radiotherapy. Complementary use of WSI and radiomics can help in the identification of EGFR mutations and their subtypes [121,122]. When adding the clinical data to radiomics and WSI, one may predict cancer treatment response or survival [123]. It has been published (patent: https://patents.google.com/patent/US11055844B2/en, accessed on 19 January 2023) that by using radiomics features of segmented cell nuclei of lung cancer it is possible to predict responses to immunotherapy (AUC up to 0.65 in the validation dataset) [124].

AI can also be used in pre-surgical evaluation. Qiu Z.B. et al., based on clinicopathological and computed tomographic texture features, established and validated a nomogram to compute the probability of invasiveness of clinical stage IA lung adenocarcinoma, which may contribute to decisions related to resection extent [125].

After surgery, AI could help in predicting prognosis. Jones G.D. et al. aimed to identify tumor genomic factors independently associated with recurrence, in the presence of aggressive, high-risk clinicopathologic variables, in patients with completely resected stages I to III lung adenocarcinoma. Further, they developed a computational ML prediction model and determined the integration of genomic and clinicopathologic features as a better predictor of the risk of recurrence, in comparison with the TNM system. The patients were identified as suitable for adjuvant therapy [126].

Concerning lung cancer, one cannot dismiss radiotherapy. Lewis and Kemp characterized The Cancer Genome Atlas (TCGA) datasets of 915 patient tumors with genome-scale metabolic Flux Balance Analysis models generated from transcriptomic and genomic datasets. It was possible to predict and experimentally validate metabolic biomarkers differentiating radiation-sensitive and -resistant tumors. This enabled integration of metabolic features with other multi-omics datasets into ensemble-based ML classifiers for radiation response. These multi-omics classifiers demonstrated the utility of personalized blood-based metabolic biomarkers for the prediction of cancer radiation sensitivity for individual patients [127].

Considering AI’s use in oncology, and specifically in lung cancer, of note is its broad potential applicability. When possible, one should connect available methods through radiomics, genomics, etc., and perform multi-level analysis to detect, stratify, or predict therapy response while simultaneously having in mind AI limitations.

Our department participates in the Horizon 2020 Research and Innovation Framework Program initiative in the Artificial Intelligence for Health Imaging—EuCanImage project: “European cancer imaging platform linked to biological and health data for next-generation artificial intelligence and precision medicine in oncology”. It aims to create a European scientific platform integrating radiological research with clinical data based on AI for targeted medicine in oncology. The project platform will host anonymized data sets of over 25,000 cancer patients. It will also be linked to biological repositories and individual healthcare systems. This will allow the creation of multidimensional AI tools integrating the clinical, radiological, and tissue level data with other predictors to create a patient-specific model, https://eucanimage.eu, accessed on 19 January 2023) [128]. In January the European Federation for Cancer Images EUCAIM project started and is the cornerstone of the Cancer Imaging Initiative, https://www.eibir.org/projects/eucaim/, accessed on 19 January 2023) [129].

Poland is one of the European countries where a lung cancer screening scientific and clinical experts’ group is very active. In Polish Pilot Lung Cancer Screening Program [20,130], the cloud platform with artificial intelligence algorithms was implemented, making it possible to connect all participating hospitals. It enables tracking the progress of the Program in real time. The algorithm makes a triage of cases. The radiologist immediately has the information from which the scans and algorithm detected a suspicious finding.

Except for low-dose CT scanning, radiomic and other models are being developed. Not all of them use AI, but the conducted studies are complementary in the studied field. For example, disease risk prediction models have been tested on large cohorts using lung cancer screening participants’ data. It was shown that lung cancer screening enrollment based on the risk prediction models is superior to NCCN Group 1 selection criteria. Clinically significant reduction of screenees with a comparable proportion of detected lung cancer cases have been observed [131].

The serum metabolome is a promising source of molecular biomarkers. They could potentially support the early detection of lung cancer. Widłak et al. identified a hypothetical metabolite-based biomarker for early detection of lung cancer; however, it requires adjustment to lifestyle-related confounding factors that putatively affect the composition of serum metabolome [132]. Smolarz et al. studied molecular components of extracellular vesicles present in serum (sEV) as non-validated potential biomarkers of lung cancer. They compared the lipid profiles of vesicles obtained from lung cancer screening study participants. A few lipids whose levels were different between compared groups were detected. In vesicles of cancer patients, ceramide Cer(42:1) was upregulated. High heterogeneity of lipid profiles of extracellular vesicles impaired the performance of classification models based on specific compounds. The data obtained do not support the use of the serum-derived “total” sEV metabolites as biomarkers for early lung cancer detection [133].

The other project aimed to determine the metabolic signature of early lung cancer and to propose a method for its early detection, considering the radiomic features of the CT image and the molecular profile of the serum [134]. The miRNA status has also been studied in the MOLTEST BIS (2015–2018) project which aimed to validate molecular signatures of early detection of lung cancer in the high-risk group [135].

A pilot study of a panel of serum metabolites (using GC/MS metabolomics) showed that they discriminate between cancer patients and healthy participants of lung cancer screening. A classifier with nine serum metabolites discriminated against cancer and control samples with 100% sensitivity and 95% specificity. This signature deserves further investigation in a larger cohort study [136]. Currently, the LCS Group is working on radiomic and other AI solutions in the lung cancer field. The connection of these approaches gives a higher probability of success.

Mikhael et al. have shown that an AI and deep learning model, called Sybil, can predict an individual’s future lung cancer risk after only one baseline computed tomography chest scan. The model was developed and internally validated using more than 12,000 LDCTs from the NLST and then externally validated in two separate large data sets (more than 23,000 LDCT screens). Model performance was very good in risk prediction at the beginning and after six years. Sybil can be run in real time in the background of the radiology reading station and be a second reader. The authors offered to provide it to other investigators to validate its usefulness or drawbacks [137].

We observe the extensive evolution and usefulness of AI in medicine. The combination of imaging features and clinical and laboratory data in AI models is a promising approach in the prediction of patients’ outcomes, response to therapies, and risk for adverse reactions development. There are 1200 AI-related registered clinical trials (https://clinicaltrials.gov/ct2/results?cond=&term=artificial+intelligence&cntry=&state=&city=&dist=, accessed on 19 January 2023) [138].

Most AI solutions are very specific and may not function properly in different circumstances, changed environments, or on different apparatus, scanners, or protocols. AI has great potential, but physicians must approach AI carefully. It seems to be reasonable as still, there are difficulties in the translation of AI applications into the clinic.

## 8. Conclusions

Lung cancer for decades remains an unsolved healthcare and social problem. Hope is placed in new technologies, namely AI. The desired broad implementation of the national lung cancer screening programs detecting the disease in the early stage and decreasing the mortality due to lung cancer may face technical and personal issues, such as the means of the shortage of specialized equipment, but mostly a shortage of expert thoracic radiologists available to read low-dose CT scans.

The tools used so far to outline and calculate the volume of the detected nodule are time-consuming and could be human-dependent. AI algorithms can perform fast triage of detected changes. AI allows significant productivity increase and lowers the scale of potential discrepancies in test results.

Even if artificial intelligence in radiology is still in its infancy, the fascinating thing is its rapid development. New open-source shared databases of CT scans are appearing and are used by scientific teams to develop and validate new solutions (e.g., IELCART) [139].

On the other hand, both large and small companies and start-ups are interested in creating AI-based healthcare solutions. As mentioned earlier, a variety of software devices have been approved by the Food and Drug Administration (FDA), but even more are under development.

For example, Google’s technology, the AI model developed by researchers in Mountain View, is based on a deep-learning model that can predict lung malignancies. This neural network was trained with lung cancer CT scans and performs as well as, or better than, trained radiologists. Eric Topol, MD, executive vice president of Scripps Research, and founder and director of Scripps Research Translational Institute, underlines that “the algorithms require prospective clinical validation but are certainly promising” [140].

In 2021 Kicky G. van Leeuwen et al. mapped 100 commercially available artificial intelligence software for radiology, with information about the existence of their scientific evidence [141]. The Authors created an online overview of CE-marked AI software products based on vendor-supplied specifications (www.aiforradiology.com accessed on 19 January 2023). Modality, subspeciality, main task, regulatory information, deployment, and pricing model had been listed (31% chest, 37% CT, 27% detection) [139]. Interestingly, they found that for 64/100 products, peer-reviewed evidence on their efficacy is lacking and only 18/100 showed potential clinical impact [141].

According to the report “AI In Medical Imaging Market—Global Outlook & Forecast 2022–2027”—https://www.reportlinker.com/p06288135/?utm_source=GNW, (accessed on 19 January 2023) in recent years, large companies, such as GE Healthcare and Siemens Healthineers, Thermo Fisher Scientific, General Electric (GE) Company, Koninklijke Philips, IBM Watson Health, and Paraxel strongly entered the AI in the medical imaging market by making huge investments and became the major players in the global AI in the medical imaging market. Manufacturers focus on introducing new products and continuously invest in research and development to remain competitive [142].

In 2021, Philips showcased the CT 5100 application that uses artificial intelligence at every step of CT image processing. AI-RAD companion chest CT (Siemens Healthineers) detects and highlights lung nodules and automatically detects tumor burden. In 2021, more than 30 countries approved AI in medical imaging technologies that are FDA and CE-approved. More than 20 start-ups from various regions have received funds to develop AI-based medical imaging technologies [143]. 

Concerning market dominance, North America is in first place in artificial intelligence in the medical imaging market followed by Europe where increasing collaborative research with extensive funding from the government can be observed. In the Asia-Pacific region, the industry is growing exponentially in Latin America, the Middle East, and Africa where the growth is slow. APAC has the highest growth over the forecast period 2022–2027 (CAGR—Compound Annual Growth Rate, 51.06% in the global industry) Read the full report: https://www.reportlinker.com/p06288135/?utm_source=GNW, (accessed on 19 January 2023) [143].

In summary, it is a matter of fact that AI solutions will be broadly used in healthcare as a support in different fields in the coming years. However, they need deep validation on a large enough number of cases, and the physician should always make the final decision. Moreover, the relevant legal regulations should be implemented and executed.

Societies need solutions that are using artificial intelligence in medicine due to the growing demand for healthcare and insufficient human resources, especially specialists, in this sector. Lung cancer screening seems to be one of the fields in which the use of AI requires a particularly urgent introduction. Creating trustworthy and secure AI tools for the LCS could be easier after obtaining financial support from the EU and other players. Will this problem be more widely recognized? We hope so, because the implementation of lung cancer screening programs is the only chance to reduce mortality due to the most common cancer death.

## Figures and Tables

**Table 2 cancers-15-01321-t002:** Steps of radiomic workflow with influencing technical factors.

Image Processing	Technical Factors
acquisition	scanner reproducibility mode
	matrix size respiratory motion artefacts
reconstruction	parameters/protocol algorithm slice thickness [62] plane pixel dimension [62] soft/sharp kernel [63]
segmentation	threshold discretization
	resampling filters
pre-processing	method
feature extraction	shape texture size volume heterogeneity intensity histogram
feature correlation	voxels
test/data analysis	method
clinical outcome modeling	statistical model

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
