# Peer review of "See Lung Cancer with an AI"

_cancers, 2023, doi:10.3390/cancers15041321_

Round 1

Reviewer 1 Report

Thank you for submitting your well-organized work and here are my recommendations for you;

1. I think Table1. can be summarized by other regional categories, for example, Northern, Western, Middle, South, and East europe, many countries you listed make readers frustrated. We had better to be frustrated by AI alorithms not by just simple country's names!

2. Unfortunately, "AI" you used in this hot word in the title was not shown in your work at all. Can you suggest some AI algorithms useful to your field? any codes or any links, please? 

Author Response

Dear Reviewer, please find below the answers to your comments. Thank you. Best regards, Joanna Bidzińska

Comment 1: Thank you for submitting your well-organized work and here are my recommendations for you.

Reply 1: We would like to thank the Reviewer for the overall positive feedback on our paper.

Thank you for the comments. Below follows a point-wise reply to the Reviewer’s comments.

Comment 2: I think Table 1. can be summarized by other regional categories, for example, Northern, Western, Middle, South, and East Europe, many countries you listed make readers frustrated. We had better be frustrated by AI algorithms, not by just simple country names!

Reply 2: We agree that Table 1 is extensive, however, we strongly believe that this is an advantage, and the information contained therein is valuable and difficult to access. The data in Table 1 is presented in a systematic form, countries are listed in alphabetical order. Unfortunately, because in each of the European countries, the lung cancer screening status is/may be individual, it is not possible to block information and provide it, for example, for the region, because there are no trends, and the situation is country-specific however we compacted it by the LCS status and hope it will be more readable.

Currently in Europe LCS is introduced in Croatia. The recently launched Lung Cancer Policy Network initiative is seeking to extend this intervention. Due to many country-specific variables, the implementation process is very complicated and is not regionally unified. Therefore, it is important to show the LCS status in each country separately. This will allow the continuation of activities depending on the current situation in the given country.

In conclusion, please see the changed Table 1 below and in the manuscript for the details.

Changes in the text, paragraph 2.2.: Table 1 has been changed accordingly

Country

Status

Albania, Bosnia and Herzegovina, Bulgaria, Cyprus, Iceland, Latvia, Lithuania, Malta, Moldova, Montenegro, North Macedonia,

Portugal, Slovenia

Unknown status. No further information available.

Belarus, Norway,

Switzerland

Implementation research. No further information available.

Austria

Implementation research. The Austrian Society of Pneumology and the Austrian Radiological Society recommended the implementation of LDCT screening and formed a taskforce focused on this in 2018. In February 2021, it announced plans to implement future pilots looking at the feasibility of LDCT screening in Austria. No further information is available.

Belgium

Implementation research. Belgium participated in a large clinical effectiveness RCT (NELSON), and results were reported in 2020. The decision to implement LCS is governed by the Flemish, Walloon or Brussels’ Ministers of Health. Funding was recently approved for a pilot in the Flemish region.

Croatia

Implementation roll-out. Croatia launched a national LDCT LCS in October 2020, as the first country in Europe. It is financed by the Croatian Health Insurance Fund. First term will conclude in 2023 before renewal.

Czech Republic

Implementation roll-out. In January 2022, a five-year national LCS was initiated by the Czech Ministry of Health. It is fully covered by public health insurance. The anticipated completion is 2026. It is expected to be renewed.

Denmark

Unknown status. Denmark previously conducted a large clinical effectiveness RCT (the DLCST), which ended in 2015. In January 2021, a proposal was submitted to the Danish National Board of Health (NBH) to introduce a screening pilot in Denmark. Currently, the NBH and Ministry of Health are considering the proposal and possible funding of a pilot as part of a HTA evaluation, which would serve as the first step towards national implementation.

Estonia

Implementation research. An ongoing regional pilot in Tartu aims to assess the feasibility of introducing a national LCS. The pilot builds on earlier implementation research, including a study conducted by the National Institute for Health Development (TAI).

Finland

Unknown status. In November 2021, a working group appointed by the Finnish Medical Association Duodecim, the Finnish Lung Doctors' Association and the Finnish Society for Oncology reviewed its clinical guidelines for lung cancer. It concluded that further research was required around the optimal eligibility criteria, interval, and cost-effectiveness of LDCT LCS before its implementation.

France

Implementation research. Pilots ongoing in France, with funding from public and private sources. In February 2022, the High Authority for Health (HAS) announced it will carry out an update to the 2016 evaluation of the evidence for LDCT LCS, and that nationally-funded implementation pilots will begin.

Germany

Implementation research. Several clinical effectiveness trials and economic evaluations have previously taken place in Germany. Two prospective evaluations performed by the Federal Office for Radiation Protection and the Institute for Quality and Efficiency in Health Care (IQWiG) were supportive of LCS in 2020, and this must be approved by The Federal Joint Committee (G-BA). Several implementation trials are ongoing (HANSE and the 4-IN-THE-LUNG-RUN).

Greece

Unknown status. As of 2021, there was no official recommendation from the Greek Ministry of Health regarding LDCT LCS. The development of a National Lung Cancer Control Strategy which includes early detection was announced. On a private level, there are implementation studies around screening programmes in a select few hospitals in Athens, Thessaloniki and Crete.

Hungary

Implementation research. The first phase of a national LDCT lung cancer screening pilot program (HUNCHEST) was completed in 2018. The second phase is currently ongoing and expected to end in 2022.

Ireland

Unknown status. There are a few small-scale opportunistic screening programmes for lung cancer, but these are exclusive to private hospitals and not endorsed by the NCCP or National Screening Service.

Italy

Formal commitment. In Italy, the Ministry of health has committed to a national programme of implementation, with a national pilot study being conducted in several centres. Italy also led several large RCTs investigating the clinical effectiveness of LDCT screening.

Luxembourg

Unknown status. In 2020, it was reported that preliminary discussions for the implementation of an LDCT LCS had begun with the National health care system and National Health Insurance Fund.

Netherlands

Implementation research. The Netherlands participated in a large clinical effectiveness RCT (NELSON); results were reported in 2020. Recruitment recently began for the Dutch arm of the 4-IN-THE-LUNG-RUN trial, which is due to end in 2024. The government has yet to support the implementation of LDCT screening.

United Kingdom

Formal commitment. The UK National Screening Committee (UK NSC) recently reviewed its recommendations for lung cancer screening, the outcome which was announced in September 2022. The UK NSC now recommends LDCT screening for current and former smokers aged 55–74. England and Scotland are currently engaged in implementation research (e.g., LUNGSCOT study and the Targeted Lung Health Check pilot programme).

Poland

Implementation roll-out. Poland initiated pilot studies for the early detection of lung cancer via LDCT screening in 2008. These pilots were introduced independently in Szczecin, Gdańsk, Poznań and Warsaw. In 2020, a national programme (the WWRP) began. The three-year centrally administered programme is being implemented. The country is divided in six macroregions with Centers of Excellence and cooperation sites.

Romania

Formal commitment. The latest NCCP has pledged that the Ministry of Health and the National Institute of Public Health will develop a national pilot programme between 2023 and 2025. The proposed target population is current and former smokers aged 50–80 who will be offered annual LDCT screening.

Serbia

Implementation research. The first screening pilot programme in Serbia is currently ongoing in the Autonomous Province of Vojvodina (APV). Initially planned as a region-wide pilot, due to the COVID-19 pandemic, it is only taking place in the South Bačka district.

Slovakia

Implementation research. A Ministry of Health taskforce is currently developing a plan for a national LDCT lung cancer screening programme in Slovakia.

Spain

Implementation research. There is no national LCS program in Spain. There have been some implementation studies. Currently, there are plans for a national pilot (CASSANDRA) led by the Spanish Society of Pneumology and Thoracic Surgery (SEPAR). This is due to begin in 2022.

Sweden

Implementation research. A regional cancer center was commissioned to deliver a regional lung cancer screening pilot that targets women in Stockholm. If successful, findings will be used to inform the development of a national organized LDCT screening programme, in which men would also be eligible to participate.

Comment 3: Unfortunately, "AI" you used in this hot word in the title was not shown in your work at all. Can you suggest some AI algorithms useful to your field? any codes or any links, please? 

Response 3: Dear Reviewer, thank you for this suggestion. Please find below examples of AI solutions (with the links) useful in the lung cancer screening field, but not only. We tested a few of them with satisfactory results and are using one daily in the assessment of lung cancer screening CT scans.

Changes in the text, paragraph 6.2-6.4:

Among computer-aided detection systems, many algorithms are useful for pulmonary nodule detection. One of the most known systems for evaluating radiological examinations based on artificial intelligence is AI-RAD Companion developed by Siemens Healthcare which allows three-dimensional segmentation of the lungs. AI-RAD companion also enables volume quantification of lungs, lobes, lesions, thoracic aorta, and coronary artery calcification. The artificial intelligence algorithms used in the AI-Rad Companion automatically use image data. It allows the automation of routine workflows of repetitive tasks. Streamlined daily work allows specialists to save time and make fewer mistakes (https://www.siemens-healthineers.com/digital-health-solutions/digital-solutions-overview/clinical-decision-support/ai-rad-companion) [70]. After the automatic assessment of the scan by AI-Rad Companion, it assists the user in the automatic interpretation of the data. In the last step, the automatically generated result can be verified, delivered, sent, and saved to the report (https://grand-challenge.org/aiforradiology/product/siemens-rad-companion-chest-ct/) [71].

Another solution had been developed by the Aidance company. Aidance provides AI-powered clinical applications for the management of lung cancer including two tools:

  1. Veye Lung Nodules – CE-certified automated lung nodule management assistant integrated into the radiological workflow. It can be used for automated lung nodule detection and quantification on chest CT scans.
  2. Veye Reporting – the interactive solution for lung nodule reporting allowing for the generation of standardized quality reporting.

Both Aidance tools can be applied in lung cancer screening but also can be used in clinical work. Aidance solutions are broadly used in the UK hospitals that are part of the Targeted Lung Health Check Program (Lung Cancer Screening) and across Europe in at least 8 countries in clinical practice. In January 2022, Aidence was acquired by RadNet, a US leader in diagnostic imaging services (https://www.aidence.com,https://www.bir.org.uk/media/477315/lung_cancer_and_ai_final_updated_v2_150622.pdf,  https://www.radnet.com) [72-74].

Aidoc’s pulmonary nodules solution is a triage and notification software indicated for use in the analysis of CT images. It flags and communicates suspected positive findings of pulmonary nodules (https://grand-challenge.org/aiforradiology/product/aidoc-pulmonary-nodules/, https://www.aidoc.com/radiology-ai/) [75,76].

Veolity is a reading platform dedicated to the lung cancer screening program for nodule detection, segmentation, and quantification. It allows nodule comparison, and report generation. It automatically marks regions that are suggestive of solid pulmonary nodules using CAD algorithm (https://grand-challenge.org/aiforradiology/product/mevis-veolity/, https://www.veolity.com) [77,78].

InferRead CT Lung is a solution for lung cancer screening. It recognizes and determines the characteristics of suspected lung nodules, provides nodules localization, size, density, and malignancy rate and report generation (https://grand-challenge.org/aiforradiology/product/infervision-ct-lung/, https://global.infervision.com) [79-80].

JLD-01K is based on the deep learning model – Convolutional Neural Network (CNN), It detects and measures the volume and diameter of nodules on CT pulmonary images. The categorization is made according to the Lung-RADS. In addition, lungs and nodules can be visualized and displayed through 3D views (https://grand-challenge.org/aiforradiology/product/jlk-inc-jld-01k/) [81].

The LCS+ tool from Coreline Soft can be used for the analysis of the 3 main lung diseases: lung cancer, COPD and coronary artery calcification based on different types of CT-scans. It enables assessment of nodule volume quantification, VDT, Lung-RADS score (https://grand-challenge.org/aiforradiology/product/coreline-soft-aview-lcs/, https://www.corelinesoft.com/en/lcs/) [82-86].

Taking into account how many solutions became available it is imortant that, before buying the system, it is usually possible to test it on cases like for example with the Contexflow https://calendly.com/contextflow-js/contextflow-demo?month=2023-02 or ask for a demo [87].

Contexflow is creating the AI tools for lung nodules assessment. In 2022 RSNA they presented an outstanding application allowing for the automatic primary nodule malignancy assessment. And showed that a ‘computer-assisted diagnosis software improved risk classification from chest CTs of screening and incidentally detected lung nodules compared with Lung-RADS’. The authors claim that ‘these results suggest the generalizability and potential clinical impact of a tool that is straightforward to implement in practice [69, 71]. Here the Contexflow company videos could be watched: ECR 2022: interviews from working radiologists about the value context flow can provide in terms of integration and diagnostic support: https://www.youtube.com/watch?v=zTR0oZEto3g, https://www.youtube.com/watch?v=VR7uNZW536E.

Forte et al., in a recent systematic review and meta-analysis of deep learning algorithms for lung cancer diagnosis, found that the pooled sensitivity and specificity of DL networks for the diagnosis of lung cancer were 93% and 68%, respectively. Current data concerning DL-based CAD tools will play an important role in LCS but still many improvements can be made [88].

6.3

High-precision detection of lung nodules is challenging. To address the problem, many groups aim to develop algorithms and methods for their automatic detection which could greatly improve work efficiency, and accuracy rate and help specialists in the diagnostic process.

Among new algorithms for the detection of pulmonary nodules are a polygonal approximation and hybrid features from CT images [91] and a neuro-evolutionary scheme [92]. For feature extraction (shape, intensity, texture) and nodule candidate classification, one can use the incorporation of 3D tensor filtering with local image feature analysis [93]. Another one of the several new methods for feature extraction and nodule candidate classification published recently is spectral analysis with the use of the optimal fractional S-transform that is applied to transform raw CT images from the spatial to time–frequency domain. After a few steps, nodule candidates are detected using rule-based and threshold algorithms in the Teager–Kaiser energy image. The proposed method exhibits a sensitivity of 97.87% and can efficiently reduce the number of false positives [94].

Zhao et al., propose a pulmonary nodule detection method based on deep learning. In the first stage, the multiscale features, and Faster R-CNN, are combined to improve the detection of small-sized lung nodules. To reduce false-positive nodules, in the second stage, a 3D convolutional neural network based on multiscale fusion was designed. The Authors report that the nodule detection model has achieved a sensitivity of 98.6%, and the proposed method reached a sensitivity of 90.5%. It has been shown that the detection method of pulmonary nodules based on multiscale fusion has a higher detection rate for small nodules. It improves the classification performance of true and false-positive pulmonary nodules. It outperforms other published methods and can help specialists in the diagnostic process [95].

Lin J. et al., developed a new 3D framework IR-UNet + + for automatic pulmonary nodule detection based on 3 steps. First is the combination of the Inception Net and ResNet as building blocks. Second, the squeeze-and-excitation structure is introduced into these building blocks for more efficient feature extraction. In the last step, two short skip pathways are redesigned based on the U-shaped network. The developed model has been shown to perform better than other lung nodule detection methods. Achieved sensitivity is 1 FP/scan, 4 FPs/scan, and 8 FPs/scan which is 90.13%, 94.77%, and 95.78%, respectively [96].

Sethy et al., developed a hybrid network for the categorization of lung histopathology images, by combining AlexNet, wavelet, and support vector machines. For pulmonary nodules classification, they feed the integrated discrete wavelet transform (DWT) coefficients and AlexNet deep features into linear support vector machines (SVMs). To train and test SVM classifiers the 5,000 digital histopathology image datasets have been used. The images were divided into three categories: normal, adenocarcinoma, and squamous carcinoma cells. An accuracy of 99.3% has been achieved using a 10-fold cross-validation method and AUC of 0.99 in classification [97].

Gugulothu et al. created a novel algorithm for early diagnosis of lung cancer detection and classification. The Adaptive Mode Ostu Binarization technique is used for the Lung Volumes isextortedas of the image with the extracted lung regions pre-processing. After the detection, segmentation takes place utilizing Geodesic Fuzzy C-Means Clustering Segmentation Algorithm. After feature extraction, nodules are classified by Logarithmic Layer Xception Neural Network Classifier. The result show enhanced classification accuracy vs prevailing techniques [98].

Despite all the knowledge we have today, there is still a great need to improve existing algorithms for lung cancer detection, diagnosis, and stage of disease assessment.

6.4

Digital pathology could be used to predict patient prognosis and response to treatment. Straight linking of specific pathological features with survival outcomes is a great challenge. Could AI complement this assessment? Could radiomics and digital pathology be interconnected?

This year for the first time Zhou et al. applied multiple machine learning algorithms, gene expression online databases, and in vitro experiments to show a potential biomarker for lung adenocarcinoma. They used 3 datasets from the omnibus database and applied R software to screen differentially expressed genes together with the immune cell infiltrates analysis. Further steps were performed with the use of multiple machine learning algorithms (least absolute shrinkage and selection operator, support vector machine-recursive feature elimination). The in vitro experiments have been also performed. The biomarker - Topoisomerase II alpha is overexpressed in lung cancer and associated with a poor prognosis [104].

Patients with the EGFR mutation have a bad prognosis. Early prediction of disease progression can help to manage the treatment course. A deep learning method using CT characteristics and clinical data to predict progression-free survival in patients with NSCLC after EGFR-TKI treatment has been developed. The combinational DeepSurv model performed better than a model based solely on clinical data and PFS can help in the prediction of tumor progression [105].

Shimada et al., have shown that CT-based radiomics with AI in predicting pathological lymph node metastasis in patients with clinical stage 0-IA non-small cell lung cancer may have broad applications such as guiding individualized surgical approaches and postoperative treatment. They used AI software Beta Version (Fujifilm Corporation, Japan), 39 AI imaging factors, including 17 factors from the AI ground-glass nodule analysis and 22 radiomic features from nodule characterization analysis [106].

The prognostic role of the stromal components in small-size tumors with lepidic components remains a challenge. Using a machine learning algorithm, the Authors analyzed the prognostic impact of each tumor component. They were able to stratify the post-operative prognosis of surgically resected stage IA adenocarcinomas and have shown that this method might help in the selection of patients with a high risk of recurrence [107].

Pan Z et al., developed an Optimized early Warning model for Lung cancer risk (OWL) using the XGBoost algorithm. They used a machine learning technique with a wide list of questionnaire-based predictors and obtained a high degree of predictive accuracy and robustness with clinical utility that can aid in screening individuals with high risks of lung cancer [108].

For the prediction of invasiveness of lung cancer based on preoperative [18F]fluorodeoxyglucose positron emission tomography and CT radiomic features 7 machine learning models were developed and validated. Radiomics features were extracted with the PyRadiomics package. The developed machine learning model was able to predict pathological highly invasive lung cancer with high discriminative ability and stability. It could be useful in quantitative risk assessment [109].

In the recent review of Ge et al., the Author provides an overview of the commonly used radiomic feature selection methods and predictive models. They compare the limitations of various methods in clinical applications. The sources of uncertainty are also presented. The impact of radiomic features, models, and methods on the integrity of radiomics studies should not be omitted [110].

Reviewer 2 Report

Radiomics in lung cancer screening, diagnosis, and prediction requires a large and highly organized professional effort, but it can be supported by artificial intelligence (AI). This article discusses whether and how AI can be used to benefit patients, radiologists, pulmonologists, thoracic surgeons, and all hospital staff who support the screening process. This is a great topic.

Radiomics Imaging has been used in cancer research and diagnosis for many years. New knowledge and continuing technological advances in the field of cancer biology have resulted in the use of multiple imaging modalities including MRI, CT, USG, PET and SPECT for screening, staging and treatment/radiation, as well as surgery, assessment of cancer response to treatment, prognosis and relapse. The addition of artificial intelligence has led to significant advances in cancer treatment. Clinicians, radiologists, and other professionals gain a trusted assistant that eases cancer patients' workload and potentially shortens their clinical decision-making time.

This paper did not review the enough literature on the intersection of AI and radiomics to analyze and compare different AI model applications in the lung cancer screening, diagnostics, and prognostication. No specific research work was presented.

Author Response

Dear Reviewer, please find below the answers to your comments. Thank you. Best regards, Joanna Bidzińska

Comment 1: Radiomics in lung cancer screening, diagnosis, and prediction requires a large and highly organized professional effort, but it can be supported by artificial intelligence (AI). This article discusses whether and how AI can be used to benefit patients, radiologists, pulmonologists, thoracic surgeons, and all hospital staff who support the screening process. This is a great topic.

Radiomics Imaging has been used in cancer research and diagnosis for many years. New knowledge and continuing technological advances in the field of cancer biology have resulted in the use of multiple imaging modalities including MRI, CT, USG, PET, and SPECT for screening, staging, and treatment/radiation, as well as surgery, assessment of cancer response to treatment, prognosis, and relapse. The addition of artificial intelligence has led to significant advances in cancer treatment. Clinicians, radiologists, and other professionals gain a trusted assistant that eases cancer patients' workload and potentially shortens their clinical decision-making time.

This paper did not review enough literature on the intersection of AI and radiomics to analyze and compare different AI model applications in lung cancer screening, diagnostics, and prognostication. No specific research work was presented

Reply 1: Dear Reviewer thank you for the comment. According to your suggestion, we added examples of the AI tools used in lung cancer screening to detect and assess the features of pulmonary nodules and beyond in paragraphs 6.2.-6.4.

Among computer-aided detection systems, many algorithms are useful for pulmonary nodule detection. One of the most known systems for evaluating radiological examinations based on artificial intelligence is AI-RAD Companion developed by Siemens Healthcare which allows three-dimensional segmentation of the lungs. AI-RAD companion also enables volume quantification of lungs, lobes, lesions, thoracic aorta, and coronary artery calcification. The artificial intelligence algorithms used in the AI-Rad Companion automatically use image data. It allows the automation of routine workflows of repetitive tasks. Streamlined daily work allows specialists to save time and make fewer mistakes (https://www.siemens-healthineers.com/digital-health-solutions/digital-solutions-overview/clinical-decision-support/ai-rad-companion) [70]. After the automatic assessment of the scan by AI-Rad Companion, it assists the user in the automatic interpretation of the data. In the last step, the automatically generated result can be verified, delivered, sent, and saved to the report (https://grand-challenge.org/aiforradiology/product/siemens-rad-companion-chest-ct/) [71].

Another solution had been developed by the Aidance company. Aidance provides AI-powered clinical applications for the management of lung cancer including two tools:

  1. Veye Lung Nodules – CE-certified automated lung nodule management assistant integrated into the radiological workflow. It can be used for automated lung nodule detection and quantification on chest CT scans.
  2. Veye Reporting – the interactive solution for lung nodule reporting allowing for the generation of standardized quality reporting.

Both Aidance tools can be applied in lung cancer screening but also can be used in clinical work. Aidance solutions are broadly used in the UK hospitals that are part of the Targeted Lung Health Check Program (Lung Cancer Screening) and across Europe in at least 8 countries in clinical practice. In January 2022, Aidence was acquired by RadNet, a US leader in diagnostic imaging services (https://www.aidence.com,https://www.bir.org.uk/media/477315/lung_cancer_and_ai_final_updated_v2_150622.pdf,  https://www.radnet.com) [72-74].

Aidoc’s pulmonary nodules solution is a triage and notification software indicated for use in the analysis of CT images. It flags and communicates suspected positive findings of pulmonary nodules (https://grand-challenge.org/aiforradiology/product/aidoc-pulmonary-nodules/, https://www.aidoc.com/radiology-ai/) [75,76].

Veolity is a reading platform dedicated to the lung cancer screening program for nodule detection, segmentation, and quantification. It allows nodule comparison, and report generation. It automatically marks regions that are suggestive of solid pulmonary nodules using CAD algorithm (https://grand-challenge.org/aiforradiology/product/mevis-veolity/, https://www.veolity.com) [77,78].

InferRead CT Lung is a solution for lung cancer screening. It recognizes and determines the characteristics of suspected lung nodules, provides nodules localization, size, density, and malignancy rate and report generation (https://grand-challenge.org/aiforradiology/product/infervision-ct-lung/, https://global.infervision.com) [79-80].

JLD-01K is based on the deep learning model – Convolutional Neural Network (CNN), It detects and measures the volume and diameter of nodules on CT pulmonary images. The categorization is made according to the Lung-RADS. In addition, lungs and nodules can be visualized and displayed through 3D views (https://grand-challenge.org/aiforradiology/product/jlk-inc-jld-01k/) [81].

The LCS+ tool from Coreline Soft can be used for the analysis of the 3 main lung diseases: lung cancer, COPD and coronary artery calcification based on different types of CT-scans. It enables assessment of nodule volume quantification, VDT, Lung-RADS score (https://grand-challenge.org/aiforradiology/product/coreline-soft-aview-lcs/, https://www.corelinesoft.com/en/lcs/) [82-86].

Taking into account how many solutions became available it is imortant that, before buying the system, it is usually possible to test it on cases like for example with the Contexflow https://calendly.com/contextflow-js/contextflow-demo?month=2023-02 or ask for a demo [87].

Contexflow is creating the AI tools for lung nodules assessment. In 2022 RSNA they presented an outstanding application allowing for the automatic primary nodule malignancy assessment. And showed that a ‘computer-assisted diagnosis software improved risk classification from chest CTs of screening and incidentally detected lung nodules compared with Lung-RADS’. The authors claim that ‘these results suggest the generalizability and potential clinical impact of a tool that is straightforward to implement in practice [69, 71]. Here the Contexflow company videos could be watched: ECR 2022: interviews from working radiologists about the value context flow can provide in terms of integration and diagnostic support: https://www.youtube.com/watch?v=zTR0oZEto3g, https://www.youtube.com/watch?v=VR7uNZW536E.

Forte et al., in a recent systematic review and meta-analysis of deep learning algorithms for lung cancer diagnosis, found that the pooled sensitivity and specificity of DL networks for the diagnosis of lung cancer were 93% and 68%, respectively. Current data concerning DL-based CAD tools will play an important role in LCS but still many improvements can be made [88].

6.3

High-precision detection of lung nodules is challenging. To address the problem, many groups aim to develop algorithms and methods for their automatic detection which could greatly improve work efficiency, and accuracy rate and help specialists in the diagnostic process.

Among new algorithms for the detection of pulmonary nodules are a polygonal approximation and hybrid features from CT images [91] and a neuro-evolutionary scheme [92]. For feature extraction (shape, intensity, texture) and nodule candidate classification, one can use the incorporation of 3D tensor filtering with local image feature analysis [93]. Another one of the several new methods for feature extraction and nodule candidate classification published recently is spectral analysis with the use of the optimal fractional S-transform that is applied to transform raw CT images from the spatial to time–frequency domain. After a few steps, nodule candidates are detected using rule-based and threshold algorithms in the Teager–Kaiser energy image. The proposed method exhibits a sensitivity of 97.87% and can efficiently reduce the number of false positives [94].

Zhao et al., propose a pulmonary nodule detection method based on deep learning. In the first stage, the multiscale features, and Faster R-CNN, are combined to improve the detection of small-sized lung nodules. To reduce false-positive nodules, in the second stage, a 3D convolutional neural network based on multiscale fusion was designed. The Authors report that the nodule detection model has achieved a sensitivity of 98.6%, and the proposed method reached a sensitivity of 90.5%. It has been shown that the detection method of pulmonary nodules based on multiscale fusion has a higher detection rate for small nodules. It improves the classification performance of true and false-positive pulmonary nodules. It outperforms other published methods and can help specialists in the diagnostic process [95].

Lin J. et al., developed a new 3D framework IR-UNet + + for automatic pulmonary nodule detection based on 3 steps. First is the combination of the Inception Net and ResNet as building blocks. Second, the squeeze-and-excitation structure is introduced into these building blocks for more efficient feature extraction. In the last step, two short skip pathways are redesigned based on the U-shaped network. The developed model has been shown to perform better than other lung nodule detection methods. Achieved sensitivity is 1 FP/scan, 4 FPs/scan, and 8 FPs/scan which is 90.13%, 94.77%, and 95.78%, respectively [96].

Sethy et al., developed a hybrid network for the categorization of lung histopathology images, by combining AlexNet, wavelet, and support vector machines. For pulmonary nodules classification, they feed the integrated discrete wavelet transform (DWT) coefficients and AlexNet deep features into linear support vector machines (SVMs). To train and test SVM classifiers the 5,000 digital histopathology image datasets have been used. The images were divided into three categories: normal, adenocarcinoma, and squamous carcinoma cells. An accuracy of 99.3% has been achieved using a 10-fold cross-validation method and AUC of 0.99 in classification [97].

Gugulothu et al. created a novel algorithm for early diagnosis of lung cancer detection and classification. The Adaptive Mode Ostu Binarization technique is used for the Lung Volumes isextortedas of the image with the extracted lung regions pre-processing. After the detection, segmentation takes place utilizing Geodesic Fuzzy C-Means Clustering Segmentation Algorithm. After feature extraction, nodules are classified by Logarithmic Layer Xception Neural Network Classifier. The result show enhanced classification accuracy vs prevailing techniques [98].

Despite all the knowledge we have today, there is still a great need to improve existing algorithms for lung cancer detection, diagnosis, and stage of disease assessment.

6.4

Digital pathology could be used to predict patient prognosis and response to treatment. Straight linking of specific pathological features with survival outcomes is a great challenge. Could AI complement this assessment? Could radiomics and digital pathology be interconnected?

This year for the first time Zhou et al. applied multiple machine learning algorithms, gene expression online databases, and in vitro experiments to show a potential biomarker for lung adenocarcinoma. They used 3 datasets from the omnibus database and applied R software to screen differentially expressed genes together with the immune cell infiltrates analysis. Further steps were performed with the use of multiple machine learning algorithms (least absolute shrinkage and selection operator, support vector machine-recursive feature elimination). The in vitro experiments have been also performed. The biomarker - Topoisomerase II alpha is overexpressed in lung cancer and associated with a poor prognosis [104].

Patients with the EGFR mutation have a bad prognosis. Early prediction of disease progression can help to manage the treatment course. A deep learning method using CT characteristics and clinical data to predict progression-free survival in patients with NSCLC after EGFR-TKI treatment has been developed. The combinational DeepSurv model performed better than a model based solely on clinical data and PFS can help in the prediction of tumor progression [105].

Shimada et al., have shown that CT-based radiomics with AI in predicting pathological lymph node metastasis in patients with clinical stage 0-IA non-small cell lung cancer may have broad applications such as guiding individualized surgical approaches and postoperative treatment. They used AI software Beta Version (Fujifilm Corporation, Japan), 39 AI imaging factors, including 17 factors from the AI ground-glass nodule analysis and 22 radiomic features from nodule characterization analysis [106].

The prognostic role of the stromal components in small-size tumors with lepidic components remains a challenge. Using a machine learning algorithm, the Authors analyzed the prognostic impact of each tumor component. They were able to stratify the post-operative prognosis of surgically resected stage IA adenocarcinomas and have shown that this method might help in the selection of patients with a high risk of recurrence [107].

Pan Z et al., developed an Optimized early Warning model for Lung cancer risk (OWL) using the XGBoost algorithm. They used a machine learning technique with a wide list of questionnaire-based predictors and obtained a high degree of predictive accuracy and robustness with clinical utility that can aid in screening individuals with high risks of lung cancer [108].

For the prediction of invasiveness of lung cancer based on preoperative [18F]fluorodeoxyglucose positron emission tomography and CT radiomic features 7 machine learning models were developed and validated. Radiomics features were extracted with the PyRadiomics package. The developed machine learning model was able to predict pathological highly invasive lung cancer with high discriminative ability and stability. It could be useful in quantitative risk assessment [109].

In the recent review of Ge et al., the Author provides an overview of the commonly used radiomic feature selection methods and predictive models. They compare the limitations of various methods in clinical applications. The sources of uncertainty are also presented. The impact of radiomic features, models, and methods on the integrity of radiomics studies should not be omitted [110].

Reviewer 3 Report

In this review, authors systematically analyzed and reviewed the necessity, current status and deficiencies of canonical low-dose computed tomography (LDCT) in lung cancer screening (LCS) program, further indicating the roles of artificial intelligence (AI) and radiomics in lung cancer management. As shown in the Introduction part, lung cancer is one of the top incidence and lethal cancers which needs early detection and therapy. Then, authors introduced the history and current developments of LCS concepts and application experiences in various medical policy systems or countries, indicating urgent requirement for containing LCS into clinical guidelines. However, shortage of screening equipments (like CT, etc) and specialists may influence the quality and efficiency of LCS. Based on decade research, AI showed its advantages in productivity improvements and cost control. In cancer diagnosis and treatment, AI is proved grand advances based on medical image analysis or multimodality analysis including gene expression profiles (like TCGA, etc), radiographic imaging and medical records databases. Finally, back to the clinical usage, AI based radiomic analytic strategies showed powerful value in lung cancer diagnosis, staging and prognostic prediction. In general, the current manuscript introduces AI’s development, pros and cons in LCS and recent clinical applications in lung cancer management, which indicating the future and challenges of radiomics. The logic and language expression are clear in this review. The clinical trials and literatures referring of this review is newly updated and solid. However, one minor point should be modified to improve the quality of this manuscript.

1.       In this manuscript, online database websites were listed with the underlines. But the websites from Line 306-311 were not. It is better to use the same format to emprise the sources of these references.

Author Response

Dear Reviewer, please find below the answers to your comments. Thank you. Best regards, Joanna Bidzińska

Comment 1: In this review, authors systematically analyzed and reviewed the necessity, current status and deficiencies of canonical low-dose computed tomography (LDCT) in lung cancer screening (LCS) program, further indicating the roles of artificial intelligence (AI) and radiomics in lung cancer management. As shown in the Introduction part, lung cancer is one of the top incidence and lethal cancers which needs early detection and therapy. Then, authors introduced the history and current developments of LCS concepts and application experiences in various medical policy systems or countries, indicating urgent requirement for containing LCS into clinical guidelines. However, shortage of screening equipments (like CT, etc) and specialists may influence the quality and efficiency of LCS. Based on decade research, AI showed its advantages in productivity improvements and cost control. In cancer diagnosis and treatment, AI is proved grand advances based on medical image analysis or multimodality analysis including gene expression profiles (like TCGA, etc), radiographic imaging and medical records databases. Finally, back to the clinical usage, AI based radiomic analytic strategies showed powerful value in lung cancer diagnosis, staging and prognostic prediction. In general, the current manuscript introduces AI’s development, pros and cons in LCS and recent clinical applications in lung cancer management, which indicating the future and challenges of radiomics. The logic and language expression are clear in this review. The clinical trials and literatures referring of this review is newly updated and solid. However, one minor point should be modified to improve the quality of this manuscript

  1. In this manuscript, online database websites were listed with the underlines. But the websites from Line 306-311 were not. It is better to use the same format to emprise the sources of these references.

Reply 1: We would like to thank the Reviewer for the overall positive feedback on our paper. Thank you for the comment.

Changes in the text 1: We unified the format of the references – all links are underlined.

Round 2

Reviewer 2 Report

After modification, this version is much better. Please add more review of the literature on the intersection of AI and radiomics to analyze and compare different AI model applications in the lung cancer screening, diagnostics, and prognostication. Please add more specific research work in the paper.

Author Response

Response letter 2                                                                                                      13.02.2023

Reviewer 2

Dear Reviewer, we appreciate your comments. Please find below the answers to your comments. Thank you.

Best regards,

Joanna Bidzińska

Comment 1: After modification, this version is much better. Please add more review of the literature on the intersection of AI and radiomics to analyze and compare different AI model applications in lung cancer screening, diagnostics, and prognostication. Please add more specific research work to the paper.

Reply 1: Dear Reviewer thank you for the comment. Accordingly, to your suggestion, we added specific research works showing the AI solutions, apart from radiomics, used in lung cancer screening/detection, diagnostics, and prognostication which enables the analysis and comparison of different AI tools applications in this field.

Changes in the text 1:

  1. Future or current directions?

Apart from radiomics, other AI tools are also used in lung cancer detection, diagnostics, and prognostication. They may and should complement each other to obtain better clinical results. Precise disease assessment is the clue for the proper clinical staging, histopathological analysis results, genomic features of the disease identification, and finally treatment of lung cancer. AI, an algorithm for the prediction and classification of objects with the use of the existing data, using for example logistic regression, can simplify and optimize these processes. Among them, we can list machine learning which includes decision trees, support vector machines (SVMs), and Bayesian networks. Further neural networks, deep learning, and convolutional neural networks can also be exploited [111,112].

For diagnosis, one can assess the nodule's histopathological features with the tool of digital pathology - whole slide imaging (WSI) which allows recognition, segmentation, classification, and markers scoring. The slide scanner transforms glass slides into digital images which are stored on the server with access for the pathologists to share expertise [113-116]. It has been shown that the WSI model can outperform the pathologist in the analysis of H&E-stained slides [117]. Moreover, it is possible to predict specific gene mutations (e.g., tumor-specific receptors) from the stained WSI slides; consequently, it enables the assessment of the treatment response and the prognosis of patients. Interestingly, expression levels seem to be related to the observed morphological features [118]. AI can also count the immune cells on slides stained for markers that are known predictor markers, like PD-L1 for immunotherapy response [119-121].

The WSI can be used also in the analysis of cytological slides. In digitalized cytology slides, the focus function is simulated through the Z-stack function and multiple layers of a different focus. What is important is the multi-potency of this method in the means of the material which can be used for the analysis. The cytological sample derived from lung cancer patients can be taken from pleural effusion, tissue aspiration, lymph node aspiration, or endobronchial ultrasound-guided fine-needle aspiration of mediastinal lymph nodes [122,123].

A combination of AI methods can help in the treatment choice including drug, surgery, and/or radiotherapy. Complementary use of WSI and radiomics can help in the identification of EGFR mutations and their subtypes [124,125]. When adding the clinical data to radiomics and WSI one may predict cancer treatment response or survival [126]. It has been published (patent: https://patents.google.com/patent/US11055844B2/en) that using radiomics features of segmented cell nuclei of lung cancer it is possible to predict responses to immunotherapy (AUC up to 0.65 in the validation dataset) [127].

AI can be used also in pre-surgical evaluation. Qiu Z.B. et al., based on clinicopathological and computed tomographic texture features established and validated a nomogram to compute the probability of invasiveness of clinical stage IA lung adenocarcinoma, which may contribute to decisions related to resection extent [128].

After surgery, AI could help in predicting prognosis. Jones G.D. et al., aimed to identify tumor genomic factors independently associated with recurrence, in the presence of aggressive, high-risk clinicopathologic variables, in patients with completely resected stages I to III lung adenocarcinoma. Further, they developed a computational ML prediction model and determined the integration of genomic and clinicopathologic features as a better predictor of the risk of recurrence, in comparison with the TNM system. The patients were identified as suitable for adjuvant therapy [129].

Concerning lung cancer, one cannot dismiss radiotherapy. Lewis and Kemp characterized The Cancer Genome Atlas (TCGA) datasets of 915 patient tumors with genome-scale metabolic Flux Balance Analysis models generated from transcriptomic and genomic datasets. It was possible to predict and experimentally validate metabolic biomarkers differentiating radiation-sensitive and -resistant tumors. This enabled integration of metabolic features with other multi-omics datasets into ensemble-based ML classifiers for radiation response.  These multi-omics classifiers demonstrated the utility of personalized blood-based metabolic biomarkers for the prediction of cancer radiation sensitivity for individual patients [130].

Considering AI's use in oncology, specifically in lung cancer, of note is its broad potential applicability. When possible, one should connect available methods through radiomics, genomics, etc., and perform multi-level analysis to detect, stratify or predict therapy response simultaneously having in mind AI limitations.

Our department participates in the Horizon 2020 Research and Innovation Framework Program initiative in the Artificial Intelligence for Health Imaging - EuCanImage project - "European cancer imaging platform linked to biological and health data for next-generation artificial intelligence and precision medicine in oncology". It aims to create a European scientific platform integrating radiological research with clinical data based on AI for targeted medicine in oncology. The project platform will host anonymized data sets of over 25,000 cancer patients. It will also be linked to biological repositories and individual healthcare systems. This will allow the creation of multidimensional AI tools integrating the clinical, radiological, and tissue level data with other predictors to create a patient-specific model, https://eucanimage.eu [131]. In January the European Federation for Cancer Images EUCAIM project started and is the cornerstone of the Cancer Imaging Initiative, https://www.eibir.org/projects/eucaim/ [132].

Poland is one of the European countries where a lung cancer screening scientific and clinical experts’ group is very active. In Polish Pilot Lung Cancer Screening Program [133, 134], the cloud platform with artificial intelligence algorithms was implemented, making it possible to connect all participating hospitals. It enables tracking the progress of the Program in real time. The algorithm makes a triage of cases. The radiologist immediately has the information in which of the scans an algorithm detected a suspicious finding.

Except for low-dose CT scanning also radiomic and other models are being developed. Not all of them use AI but the conducted studies are complementary in the studied field. For example, disease risk prediction models have been tested on large cohorts using lung cancer screening participants' data.  It was shown that lung cancer screening enrollment based on the risk prediction models is superior to NCCN Group 1 selection criteria. Clinically significant reduction of screenees with a comparable proportion of detected lung cancer cases have been observed [135].

The serum metabolome is a promising source of molecular biomarkers. They could potentially support the early detection of lung cancer. Widłak et al. identified a hypothetical metabolite-based biomarker for early detection of lung cancer however it requires adjustment to lifestyle-related confounding factors that putatively affect the composition of serum metabolome [136]. Smolarz et al. studied molecular components of extracellular vesicles present in serum (sEV) as non-validated potential biomarkers of lung cancer. They compared the lipid profiles of vesicles obtained from lung cancer screening study participants. A few lipids whose levels were different between compared groups were detected.  In vesicles of cancer patients, ceramide Cer(42:1) was upregulated. High heterogeneity of lipid profiles of extracellular vesicles impaired the performance of classification models based on specific compounds. The data obtained do not support the use of the serum-derived “total” sEV metabolites as biomarkers for early lung cancer detection [137].

The other project aimed to determine the metabolic signature of early lung cancer and to propose a method for its early detection, considering the radiomic features of the CT image and the molecular profile of the serum [138]. The miRNA status also has been studied in the MOLTEST BIS (2015-2018) project which aimed to validate molecular signatures of early detection of lung cancer in the high-risk group [139].

A pilot study of a panel of serum metabolites (using GC/MS metabolomics) showed that they discriminate between cancer patients and healthy participants of lung cancer screening. A classifier with nine serum metabolites discriminated against cancer and control samples with 100% sensitivity and 95% specificity. This signature deserves further investigation in a larger cohort study [140]. Currently, the LCS Group is working on radiomic and other AI solutions in the lung cancer field. The connection of these approaches gives a higher probability of success.

Mikhael et al., have shown that an AI and deep learning model, called Sybil, predict an individual's future lung cancer risk after only one baseline computed tomography chest scan. The model was developed and internally validated using more than 12,000 LDCTs from the NLST and then externally validated in two separate large data sets, (more than 23,000 LDCT screens). Model performance was very good in risk prediction at the beginning and after 6 years. Sybil can be run in real-time in the background of the radiology reading station and be a second reader. The authors offered to provide it to other investigators to validate its usefulness or drawbacks [141].

We observe the extensive evolution and usefulness of AI in medicine. The combination of imaging features, and clinical and laboratory data in AI models is a promising approach in the prediction of patients’ outcomes, response to therapies, and risk for adverse reactions development. There are 1200 AI-related registered clinical trials (https://clinicaltrials.gov/ct2/results?cond=&term=artificial+intelligence&cntry=&state=&city=&dist=) [142].

Most AI solutions are very specific and may not function properly in different circumstances, changed environments, or on different apparatus, scanners, or protocols. AI has great potential but physicians approach AI carefully. It seems to be reasonable as still, there are difficulties in the translation of AI applications into the clinic.